# Evidence That DDR1 Promotes Oligodendrocyte Differentiation during Development and Myelin Repair after Injury

**DOI:** 10.3390/ijms241210318

**Published:** 2023-06-19

**Authors:** Ruyi Mei, Wanwan Qiu, Yingying Yang, Siyu Xu, Yueyu Rao, Qingxin Li, Yuhao Luo, Hao Huang, Aifen Yang, Huaping Tao, Mengsheng Qiu, Xiaofeng Zhao

**Affiliations:** 1Zhejiang Key Laboratory of Organ Development and Regeneration, College of Life and Environmental Sciences, Hangzhou Normal University, Hangzhou 311121, China; meiry@ibmc.ac.cn (R.M.); yangyingying@stu.hznu.edu.cn (Y.Y.); chouwanwan@stu.hznu.edu.cn (W.Q.); 2021210315087@stu.hznu.edu.cn (S.X.); 2021210315225@stu.hznu.edu.cn (Y.R.); 2021210315016@stu.hznu.edu.cn (Q.L.); 2022011010021@stu.hznu.edu.cn (Y.L.); 20173007@hznu.edu.cn (H.H.); 20108025@hznu.edu.cn (A.Y.); 20118044@hnzu.edu.cn (H.T.); 20099003@hznu.edu.cn (M.Q.); 2Hangzhou Institute of Medicine (HIM), Chinese Academy of Sciences, Hangzhou 310022, China

**Keywords:** *Ddr1*, oligodendrocyte, differentiation, myelination, ERK

## Abstract

Oligodendrocytes generate myelin sheaths vital for the formation, health, and function of the central nervous system. Mounting evidence suggests that receptor tyrosine kinases (RTKs) are crucial for oligodendrocyte differentiation and myelination in the CNS. It was recently reported that discoidin domain receptor 1 (Ddr1), a collagen-activated RTK, is expressed in oligodendrocyte lineage. However, its specific expression stage and functional role in oligodendrocyte development in the CNS remain to be determined. In this study, we report that *Ddr1* is selectively upregulated in newly differentiated oligodendrocytes in the early postnatal CNS and regulates oligodendrocyte differentiation and myelination. *Ddr1* knock-out mice of both sexes displayed compromised axonal myelination and apparent motor dysfunction. *Ddr1* deficiency alerted the ERK pathway, but not the AKT pathway in the CNS. In addition, *Ddr1* function is important for myelin repair after lysolecithin-induced demyelination. Taken together, the current study described, for the first time, the role of *Ddr1* in myelin development and repair in the CNS, providing a novel molecule target for the treatment of demyelinating diseases.

## 1. Introduction

In the central nervous system (CNS), myelin sheaths, the major component of the white matter, are elaborated by mature oligodendrocytes (OLs). Myelin plays a crucial role in the rapid and precise conduction of action potentials [1,2,3,4]. Increasing evidence has demonstrated that abnormal myelin development contributes to the disease pathogenesis including Alzheimer’s disease, major depression, and schizophrenia [5,6,7].

The progression of OL lineage cells is controlled by both intrinsic mechanisms and environmental factors. Extracellular matrix (ECM) proteins play a critical role in the development and maintenance of the nervous system. A multitude of studies have revealed ECM molecules can regulate neuronal and glial migration, memory storage, myelination, blood−brain barrier formation and maintenance, synaptogenesis, and so on [8]. Collagens, the major components of the ECM in the environment surrounding neurons and glial cells, regulate oligodendrocyte differentiation and myelination [9,10,11]. Collagen IV, one of the most abundant types of collagens in the brain, binds to discoidin domain receptor 1 (Ddr1), a receptor tyrosine kinase (RTK) [12]. Previously, *Ddr1* was shown to be enriched in the murine transcriptome of OL lineage cells and post-mortem samples of human cerebral cortex [13,14]. RNA in situ hybridization revealed that expression of *Ddr1* follows the progression of myelination [15]. Based on these and other studies, Vilella et al. described that *Ddr1* is highly expressed in cells of oligodendrocyte lineage [16]. However, the functional involvement of Ddr1 in oligodendrocyte development and myelinogenesis has not been investigated.

In this study, we report that *Ddr1* is selectively upregulated in oligodendrocytes during differentiation and myelin formation stages. Functional studies with *Ddr1* knock-out mice demonstrated that *Ddr1* deficiency causes a developmental delay of oligodendrocyte differentiation in the CNS, resulting in a reduced number of myelinated axons. Moreover, we demonstrated that *Ddr1* deficiency attenuates myelin regeneration after demyelinating injury.

## 2. Results

### 2.1. Ddr1 Is Selectively Up-Regulated in Newly Differentiated Oligodendrocytes

To characterize the stage specificity of *Ddr1* expression in the CNS in detail, we investigated the expression of *Ddr1* in the developing mouse brain. RNA in situ hybridization (ISH) revealed that *Ddr1*-positive cells started to emerge in the corpus callosum (CC) (black arrowheads) at around postnatal day 7 (P7) (Figure 1A–C), increased progressively (Figure 1D), peaked at around P15 and then declined gradually thereafter (Figure 1E,F). A similar expression pattern was detected in the developing spinal cords (Appendix A). The spatiotemporal pattern of *Ddr1* expression suggests its selective up-regulation in newly differentiated oligodendrocytes (Figure 1A–D and Appendix A).

The selective expression of *Ddr1* in differentiated OLs was further verified in the spinal cords of *Cnp^cre/+^*;*Nkx2.2^fl/fl^* mice [17,18]. As previously reported, *Cnp^cre/+^*;*Nkx2.2^fl/fl^* conditional mutant mice delayed the differentiation of oligodendrocyte precursor cells (OPCs) in the spinal cord [19]. Consistent with the idea that *Ddr1* is expressed by differentiatiated OLs, the number of *Ddr1*^+^ cells was dramatically reduced in *Nkx2.2* conditional knock-out mice as compared to in the control group (Appendix A).

The selective expression of *Ddr1* in differentiated Ols was also validated by its co-expression with various well-established oligodendrocyte markers in P15 brain tissues. As shown in Figure 1G–I, the majority of *Ddr1*-positive cells were co-stained with MYRF, a transcription factor that is specifically transcribed in OL cells at the onset of differentiation [20]. Only a small percentage of *Ddr1*^+^ cells were CC1^+^, but not ASPA^+^ mature oligodendrocytes in the corpus callosum [21,22,23] (Figure 1G–I). In addition, *Ddr1*^+^ cells did not co-express the astrocyte lineage marker GFAP, the microglial marker IBA1, as well as the neuronal marker NeuN in the corpus callosum at P15 (Figure 1J–L). Together, these data manifested that *Ddr1* is transiently up-regulated in the newly differentiated OLs in early postnatal CNS, suggestive of its important role in OL maturation and myelination.

### 2.2. Ddr1 Mutation Results in a Transient Delay of OL Differentiation

To elucidate the in vivo function of *Ddr1* in OL differentiation and myelination, we examined the expression of several well-documented molecular markers for differentiated OLs in *Ddr1* knock-out mice (*Ddr1*-KO). Double immunofluorescent staining revealed that the percentage of CC1^+^ cells in OLIG2^+^ OLs was significantly decreased in the corpus callosum from P10 to P15; however, a comparable percentage of CC1^+^ OLs was observed in the P30 corpus callosum (Figure 2A–D). Similar results were obtained for the expression of MYRF (Figure 2E–H). In parallel, *Mbp* and *Plp1* expression levels in the corpus callosum by ISH were attenuated in *Ddr1*-KO at P10 and P15, but not at P30 (Figure 2I–N’,R). Consistently, the number of *Pdgfrα*^+^ OPCs increased markedly in the mutants at P10 and P15, but no significant difference was observed at P30 (Figure 2O–Q’,S). Together, these results demonstrated that loss of *Ddr1* expression significantly delayed OL differentiation and myelin genes expression in the developing central nervous system.

### 2.3. Deficiency of Ddr1 in OL Impairs Axonal Myelination in the CNS

The early effects of *Ddr1* mutation on OL differentiation and myelin genes expression in the CNS have suggested its important role in myelination. To investigate this possibility, we examined the development of myelin tracts by Black Gold II staining, which revealed a thinner corpus callosum in the mutants (Figure 3). Moreover, the density of white matter fiber tracts in the *Ddr1*-KO corpus callosum was clearly reduced at P15 (Figure 3B,B’).

Previous studies demonstrated that the callosal thickness is influenced by varying degrees of axonal myelination, redirection, and pruning [24]. Therefore, we next investigated axonal myelination by TEM in control and *Ddr1* mutants at different developmental stages. It was found that the percentage of myelinated axons in *Ddr1*-KO mice was significantly reduced at P15 and P30 (Figure 4A–B’,D). The proportion of small-diameter axons (<0.5 μm) was smaller in *Ddr1*-KO mice (Figure 4H,I). Meanwhile, the myelin thickness was thinner in *Ddr1*-KO littermates, as indicated by the larger average g-ratios of myelin sheaths (Figure 4E,F). Moreover, the reduced percentage of myelinated axons and thinner myelin sheaths were also observed in the two-month-old adult mutant mice (Figure 4C–C’,D,G). At this stage, the proportion of small-diameter axons increased in knock-out mice, while the number of large-diameter axons decreased (Figure 4J). Based on these observations, we concluded that ablation of *Ddr1* impairs axonal myelination in the CNS, leading to changes in callosal thickness.

### 2.4. Ablation Ddr1 Leads to an Abnormal Motor Function

Myelination in the central nervous system is critical for regulating motor functions [25,26]. We next tested whether a de novo myelin deficit could cause maladaptive behaviors in young adult mice. First, we chose the open field test, a non-conditioned procedure commonly used for assessing locomotor activity in rodents (Figure 5A,A’). In this test, *Ddr1*-KO mice were hypoactive, and the total distance traveled during a 30-minute test was markedly decreased (Figure 5B). In addition, young adult *Ddr1*-KO mice showed a lower ratio of the travel distance within the central area to the total travel distance than wild-type controls (Figure 5C). To further analyze the motor coordination capabilities of mutants, rotarod testing was performed. Noticeably, *Ddr1*-KO mice showed apparently shorter latency to fall than wild-type mice (Figure 5D). The significant decline in the ability to sustain movement on a rotating rod indicated a lack of motor coordination in *Ddr1*-KO mice. Finally, we used the grip strength to measure the forelimb peak force in mutant and control mice, and we found that *Ddr1* mutation caused a significant decrease in grip strength (Figure 5E). Collectively, these behavior tests indicated that deletion of the *Ddr1* gene leads to the motor dysfunction.

### 2.5. Ddr1 Is Critical for Remyelination in the CNS

As *Ddr1* deletion has a significant impact on oligodendrocyte differentiation and early myelinogenesis, we next investigated the function of *Ddr1* in axonal remyelination following demyelinating injury. Axonal demyelination was induced in the mice by injecting lysolecithin (LPC) into the corpus callosum at P60 (Figure 6A). The mice were analyzed at 7, 14, and 21 days post-injection (dpi), corresponding to the stages of OPC recruitment, differentiation, and remyelination, respectively (Figure 6B). Axonal demyelination was confirmed by the marked reduction of myelin basic protein (MBP) staining in the corpus callosa of both wild-type and *Ddr1*-KO mice at 7 dpi (Figure 6C,C’). Interestingly, the density of differentiated CC1^+^ OLs in the lesion sites markedly decreased in the *Ddr1*-KO mice than in the wild-type mice, suggesting the reduced formation of new mature OLs in the demyelinating tissues (Figure 6F–F’,I). Remyelination was severely hampered in *Ddr1* mutants, as evidenced by apparent reduction of MBP and CC1 expression in lesions at both 14 dpi and 21 dpi (Figure 6D–E’,G–H’,I). Together, these results indicated that ablation of *Ddr1* inhibits myelin repair in demyelinated lesions induced by LPC.

### 2.6. Ddr1 Might Enhance the Differentiation of Oligodendrocytes by Restraining ERK Signaling

ERK is a crucial signaling molecule for OL differentiation and axonal myelination [27,28]. *Ddr1* has been reported to affect intracellular signaling through the AKT and ERK pathways in prostate cancer cells [29,30]. We therefore assessed whether *Ddr1* influences ERK and AKT protein phosphorylation in the CNS and therefore OL differentiation and myelination. The Western blot analysis showed that the ratio of phosphorylated ERK to total ERK was dramatically increased in the brainstem of *Ddr1*-KO mice at P10 and P15, while the level of MBP was significantly reduced (Figure 7A–C,E). Nevertheless, the downstream AKT in the brainstem was not altered (Figure 7A,D,F). These results suggest that *Ddr1* may enhance OL differentiation and myelination by attenuating the MAPK/ERK signaling pathway.

## 3. Discussion

The identification of factors regulating myelination is critical to the development of therapeutic treatment of axonal hypomyelination and demyelination of the CNS. Axonal myelination is a highly complex process that requires coordination of multiple pathways such as Sonic hedgehog, receptor tyrosine kinases (RTKs), or Wnt/β-catenin [31,32,33,34,35]. Although *Ddr1* was shown to be enriched in oligodendroglia, little was known about its in vivo role in oligodendrocyte development. In this study, for the first time, we provided the genetic and molecular evidence that *Ddr1* regulates the tempo of OL differentiation and myelination and influences the motor function of animals.

It was previously reported that *DDR1* is detected in the soma of OLs in human CNS [14]. In the present study, we determined that *Ddr1* is predominantly expressed in the newly differential OLs in the early postnatal CNS when oligodendrocytes undergo active differentiation and myelination in mice (Figure 1 and Appendix A). The strong expression of *Ddr1* in differentiated OLs appeared to be important for regulating OL differentiation and myelin formation. In support of this idea, *Ddr1* deficiency led to a transient developmental delay of OL differentiation and myelin genes expression (Figure 2). Unexpectedly, the callosal thickness became thinner in the mutants (Figure 3), possibly due to the delayed OL differentiation and a reduced degree of axonal myelination in the mutants. TEM analyses revealed fewer myelinated axons and a reduced number of small-diameter (<0.5 μm) myelinated axons (Figure 4). Since myelin is key to axonal signal transduction and related motor function in the CNS [36,37], *Ddr1*-deficient mice indeed displayed motor dysfunction (Figure 5). Interestingly, in the open field test, the mice not only reduced their total travelled distance, but also avoided the center of the field (Figure 5), which is also associated with thigmotaxis or anxiety-related behaviors in rodents. Considering that *Ddr1* mutation resulted in a transient delay of OL differentiation (Figure 2) and impairment oligodendroglia maturation affects glutamatergic neuron function consequently causing anxiety-related behaviors in mice [38], it remains plausible that deficiency of *Ddr1* in OLs can also lead to anxiety-like behaviors. Although we have demonstrated that *Ddr1* was not expressed in NeuN^+^ neurons (Figure 1), generation of conditional mutants is necessary for dissecting out the cell-autonomous function of *Ddr1* in OL development and myelination in the future.

Previous work suggested that *Ddr1* is upregulated in remyelinating oligodendrocytes [39], but its role in the remyelination process was not defined. In this study, we examined the function of *Ddr1* in the LPC-induced demyelination and found that the proportion of mature oligodendrocytes (MBP^+^/CC1^+^) was significantly decreased in the lesions of *Ddr1*-KO mice. These results showed that ablation of *Ddr1* significantly aggravated the demyelination injury and inhibited axonal remyelination in vivo (Figure 6).

ERK is a key component of the RAS/RAF/MEK/ERK signaling pathway that regulates cell proliferation and differentiation. The complete picture of how the ERK proteins regulate myelination is still unclear, and conflict results have been reported. A number of studies demonstrated increased levels of ERK1/2 activity in mature OLs induced by growth factors, which decrease myelin protein levels and inhibit the OL differentiation [40,41,42], and inhibition of ERK1/2 signaling dramatically promotes OPC differentiation and OL maturation [27]. Conversely, other studies have reported that activation of ERK1/2 significantly increases myelination [43,44]. One possible explanation for the difference between these conflict results might be the different developmental stages of OLs studied. In our study, we demonstrated that *Ddr1* was selectively up-regulated in newly differentiated OLs (Figure 1 and Appendix A), and *Ddr1* was required to suppress ERK signaling for OL differentiation (Figure 7). A more recently identified negative regulators of oligodendrocyte differentiation is the MAPK/ERK pathway [27,45,46,47]. It was previously demonstrated that *Ddr1* could modulate the activity of the AKT/ERK pathway and plays an important role in the migration and adhesion of cancer cells [30,48,49]. The Western blot analysis confirmed that inhibition of *Ddr1* upregulates the ratio of p-ERK to total ERK without altering the AKT signaling pathway (Figure 7). Therefore, we proposed that *Ddr1* deletion leads to the abnormal CNS myelin malfunctions and motor dysfunction possibly through an ERK-dependent mechanism. As *Ddr1* exerts a positive effect on the remyelination process, it could be developed as a potential drug target for promoting myelin repair in neurodegenerative diseases.

## 4. Materials and Methods

### 4.1. Animals

*Ddr1*-KO mice were purchased from Lexicon Pharmaceuticals Incorporated (LEXKO-1928) and have been previously described [50,51,52]. The mice were housed in a regulated temperature environment (22 ± 1 °C) in a 12 h light/dark cycle, with ad libitum access to food and water. The general health status of the experimental mice was analyzed by daily observation. The mice were sacrificed under deep anesthesia induced by Avertin. *Nkx2.2*^flox^ and *Cnp*-Cre mouse lines were described previously [17,53]. All animal experiments were conducted in mice of both genders except for behavior tests. In this study, all animal protocols followed ethical guidelines and were approved by the Laboratory Animal Center, Hangzhou Normal University, and the Animal Ethics Committee of Hangzhou Normal University, China (permit number: 2022-1063; approved on 3 March 2022).

### 4.2. In Situ RNA Hybridization (ISH)

Brain sections were collected consecutively from Bregma 1.1 mm to Bregma 1.34 mm of the mouse brains at various developmental stages. Tissues were fixed with 4% paraformaldehyde (PFA) in PBS (pH 7.4) at 4 °C overnight. Tissues were then cryoprotected in 30% sucrose, embedded in an optimal cutting temperature compound (OCT) medium and sectioned on a cryostat at 18 µm. The procedures for in situ hybridization (ISH) have been described previously [19]. The digoxin-labeled RNA probes used for ISH corresponded to nucleotides 1210-2178 of mouse *Plp1* mRNA (NM_011123.4), nucleotides 1485-2453 of mouse *Mbp* mRNA (NM_010777.3), nucleotides 2870-3678 of mouse *Ddr1* mRNA (NM_007584.3), and nucleotides 1613-2463 of mouse *Pdgfrα* mRNA (NM_01158.3).

### 4.3. Immunofluorescence Staining

Brain sections were collected consecutively from Bregma 1.1 mm to Bregma 1.34 mm of the mouse brains at the corresponding stages. Animals were fixed by transcardial perfusion with cold 4% PFA, after the animals were deeply anesthetized. Tissues were then isolated and post-fixed overnight, cryoprotected in 30% sucrose, embedded in a frozen section medium (6502; Thermo Scientific, Waltham, MA, USA) and sectioned into 16 µm on a cryostat. Experimental procedures for immunofluorescence were described previously [46,54]. Briefly, the sections were washed with PBS and then incubated in a citrate antigen retrieval solution (pH 6.0). The sections were then rinsed three times in PBS, blocked with 10% goat serum in PBS with 0.2% Triton-X-100 for 1 h and immediately incubated with primary antibodies in a blocking solution at 4 °C overnight. The sections were washed three times in PBS and incubated with secondary antibodies at room temperature for 2 h and then washed and mounted in Mowiol mounting medium (MMM) with 4′,6-diamidino-2-phenylindole (DAPI). Fluorescent images were collected by a Nikon epifluorescence microscope. The primary antibodies used were as follows: anti-OLIG2 (1:500, OB-PRB009, Oasis Biofarm, Hangzhou, China), anti-CC1 (1:200, OB-PGP027, Oasis Biofarm, Hangzhou, China), anti-ASPA (1:100, OB-PRB037,Oasis Biofarm, Hangzhou, China), anti-NeuN (1:500, OB-PRB039, Oasis Biofarm, Hangzhou, China), anti-SOX10 (1:500, OB-PGP001, Oasis Biofarm, Hangzhou, China), anti-MYRF (1:500, OB-PRB007, Oasis Biofarm, Hangzhou, China), anti-IBA1 (1:500, OB-PRB029, Oasis Biofarm, Hangzhou, China), anti-MBP (1:100, MAB382, Millipore, United States), and anti-GFAP (1:500, OB-PRB005, Oasis Biofarm, Hangzhou, China). The secondary antibodies used were Alexa Fluor 488/594-conjugated antibodies (Invitrogen, Carlsbad, CA, USA).

### 4.4. Transmission Electron Microscopy (TEM)

The brain sections were collected consecutively from Bregma 1.1 mm of the mouse brains from P15 to P60. Mice were perfused with a phosphate buffer solution containing 2.5% glutaraldehyde and 4% PFA. The corpus callosum tissues were isolated and post-fixed in 1% osmium tetroxide for 1 h. The tissues were then washed in a 0.1 M cacodylate buffer, dehydrated in graded ethanol and embedded in epoxy resins. Ultrathin sections (0.5 µm) were stained with toluidine blue and observed under a transmission electronic microscope.

### 4.5. Black Gold II Staining

The brain sections were collected consecutively from Bregma 1.1 mm to Bregma 1.34 mm of the P60 mouse brains or of the P15 and P30 mouse brains correspondingly. Brain tissues from P15 to P60 were fixed in 4% PFA and cut at 50 μm on a freezing sliding microtome. In addition, experimental procedures for Black Gold II staining were described previously [55,56] Tissue sections were hydrated and incubated in a 0.3% Black Gold II solution (AG105, Millipore, Kenelworth, NJ, USA ) at 60 °C for 12 min. If myelin impregnation was not complete under the microscope, the sections were returned to the staining solution until the finest myelinated fibers turned to black. After be further washed in distilled water, the sections were fixed in 1% sodium thiosulfate for 3 min. The sections were then rinsed in water, dehydrated in alcohols and coverslipped with a mounting medium.

### 4.6. Induction of White Matter Demyelination

Lysolecithin (LPC, L1381; Sigma, St. Louis, MO, USA) was dissolved in PBS at a final concentration of 1% (g/ml). P60 mice were anesthetized with Avertin by intraperitoneal injection and demyelination was induced by the focal injection of 2 μL 1% lysolecithin into the corpus callosum (1.00 mm posterior to bregma, 1.04 mm lateral to bregma, and 2.2 mm deep). LPC was delivered at a rate of 0.075 μL/min, and the needle remained in position for 10 min after LPC delivery and then was pulled out slowly. The wound was sutured, and the mice were placed on a heating pad until woken up.

### 4.7. Western Blot

Brainstem tissues were isolated and homogenized. Homogenates in a lysis buffer (R0278, Sigma, St. Louis, MO, USA) with a protease inhibitor cocktail (P8340; Sigma, St. Louis, MO, USA). Lysates were centrifuged at 12,000× *g* and 4 °C for 15 min to get rid of the unsolved debris. The concentration of the supernatant was measured by a BCA assay (23225, Thermo Scientific, Waltham, MA, USA). Proteins in samples were separated by 6–12% SDS-PAGE, transferred to a Immobilon-P Transfer Membrane (Millipore, Kenelworth, NJ, USA) and then incubated with indicated primary antibodies diluted in a blocking buffer at 4 °C overnight after blocking by a 5% non-fat milk solution in TBST (50 mM Tris, pH 7.4, 150 mM NaCl, and 0.1% Tween 20) for 1 h at RT. The primary antibodies used were as follows: anti-ERK1/2 (1:5000, ab184699; Abcam), anti-Phospho-ERK1/2 (1:5000, ab76299; Abcam), anti-ACTB (1:10,000, AC026; ABclonal, Wuhan, China), anti-MBP (1:2000, MAB382; Millipore, Kenelworth, NJ, USA), anti-AKT1 (1:5000, A17909; Abclona,), and anti-Phospho-AKT1^S473^ (1:5000, ab81283; Abcam, Cambridge, MA, USA). Protein detection was achieved with an enhanced chemiluminescence system (Amersham Biosciences, NA, UK).

### 4.8. Behavior Assays

All behavior experiments were performed during the standard light phase, usually between 9:00 a.m. and 2:00 p.m. In all tests, the cages containing mice were moved into the testing room 1 h before the first trial began. The testers used mixed genotype home cages and were blinded to genotype. The apparatus was cleaned and sterilized, before and after each test to prevent bias due to olfactory cues. In order to exclude the influence of hormones, all behavioral tests were conducted on male mice at P60. The data were analyzed, after all the behavior tests were conducted. The open-field test was performed as previously described [57]. A plastic open-field chamber (30 cm × 30 cm × 34.5 cm) was used. The mice were placed in a chamber, and their activity was recorded by the tracking software EthoVision XT 12 (Noldus, PA Wageningen, Netherlands). The total distance traveled, the distance traveled in the center, and the ratio of the distance traveled in the center to the total distance traveled over 30 min were recorded during the testing period. The rotarod test was conducted as follows. To evaluate the sensorimotor coordination, the mice were placed on an accelerating rotarod (Mouse Rota-Rod NG, Harikul Science, UB47650, Ugo basile, Comerio, Italy) and assessed for the ability to maintain balance on the rotating bar that accelerated from 0 to 40 rpm over a 5 min period. The mice were tested for 4 trials on the first day with a 30 min gap between trials. The latency before falling from the rod was recorded. The forelimb grip strength assessment was shown as follows. The forelimb grip strength was measured using a grip strength apparatus (Bioseb, France). The grip strength of each mouse was measured on six trials per day, and the average peak force per animal was computed.

### 4.9. Statistical Analysis

All data were analyzed using Prism (GraphPad) and presented as mean ± SEM. The two-way analysis of variance (ANOVA) followed by a multiple comparison was used for the quantitative analysis of data over time and between genotypes. The unpaired two-tailed Student’s *t* test was used for analysis between two groups with one variable. In addition, a *p*-value of <0.05 was considered as statistically significant. For each analysis, the results from independent animals were treated as biological replicates (*n* ≥ 3). For Western blotting and staining results, statistical analyses were performed after the subtraction of the background intensity and normalization with controls in each batch of experiments to minimize the influences of batch-to-batch variations. Detailed statistical information for each experiment was included in the figure legends.

## Figures and Tables

**Figure 1 ijms-24-10318-f001:**
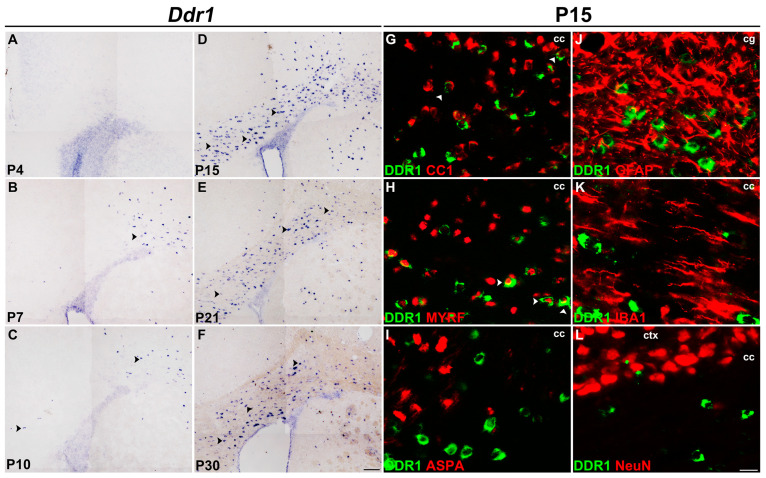
*Ddr1* was selectively up-regulated in newly differentiated OLs in the CNS. (**A**–**F**) Expression of *Ddr1* in the forebrain tissues as detected by RNA in situ hybridization (ISH). Black arrowheads highlight *Ddr1*^+^ cells. The number of *Ddr1*-positive cells in the corpus callosum drastically increased at P15 and then declined thereafter. The scale bar represents 100 µm. (**G**–**L**) Immunostaining with indicated antibodies combined with *Ddr1* mRNA ISH was performed in the corpus callosum from P15 wild-type mice. Double-positive cells are represented by white arrowheads. *Ddr1*^+^ cells were co-expressed with oligodendrocyte marker CC1 and MYRF, but they were not detected in ASPA^+^ mature oligodendrocytes, GFAP^+^ astrocytes, IBA1^+^ microglia, and NeuN^+^ neurons. cc—corpus callosum; cg—cingulum; ctx—cortex; the scale bar represents 25 µm.

**Figure 2 ijms-24-10318-f002:**
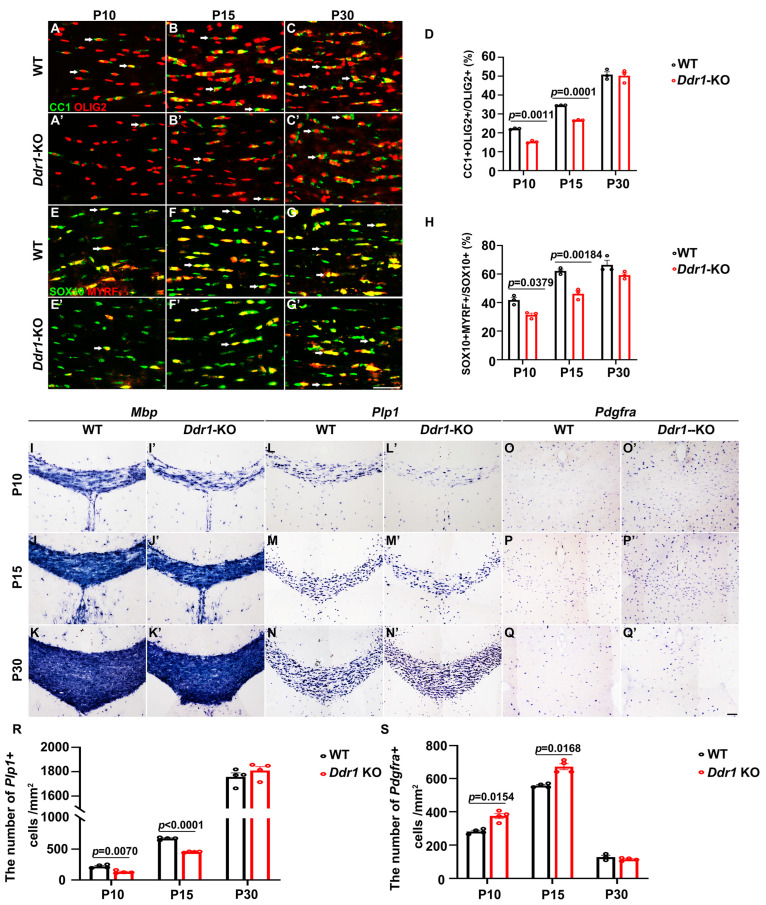
Ablation of the *Ddr1* gene delayed the differentiation and maturation of oligodendrocytes. (**A**–**C’**) Brain sections from wild-type (WT) and *Ddr1*-KO mice were subjected to anti-OLIG2 (red) and anti-CC1 (green) double immunostaining. Arrows pointed to CC1/OLIG2 double-positive cells. The scale bar represents 25 µm. (**D**) Quantitative analysis of the ratio of CC1^+^ cells in OLIG2^+^ oligodendrocytes at indicated stages. (**E**–**G’**) Double labeling of SOX10 and MYRF in P10, P15, and P30 WT and *Ddr1*-KO mice. The scale bar represents 25 µm. (**H**) Quantification of MYRF/SOX10 double-positive cells per section in the WT and *Ddr1*-KO mouse corpus callosa at indicated stages. (**I**–**Q’**) *Mbp*, *Plp1*, and *Pdgfrα* mRNA expression in coronal forebrain sections of wild-type (**I**–**Q**) and *Ddr1*-KO (**I’**–**Q’**) mice at indicated stages. (**R**,**S**) Quantification of the number of *Plp1*^+^ cells per mm^2^ and the number of *Pdgfrα*^+^ cells per mm^2^ (*n* = 3) in the corpus callosum from wild-type and *Ddr1*-KO mice at indicated stages. The scale bar represents 100 µm. Error bars indicate means ± SEM.

**Figure 3 ijms-24-10318-f003:**
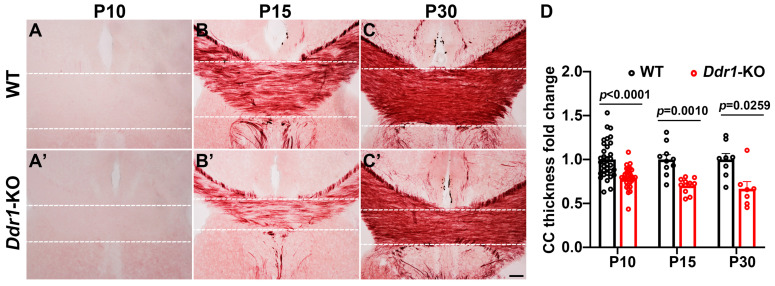
Deficiency of *Ddr1* altered the callosal thickness. (**A**–**C’**) Detection of myelin tracts within the corpus callosum at P10, P15, and P30 for wild-type mice (**A**–**C**) and *Ddr1*-KO mice (**A’**–**C’**) by Black Gold II staining. Dashed lines delineate the region of interest (ROI). The scale bar represents 100 µm. (**D**) Quantification of the corpus callosal thicknesses of wild-type and *Ddr1*-KO mice at indicated stages. Error bars indicate means ± SEM. *n* > 3.

**Figure 4 ijms-24-10318-f004:**
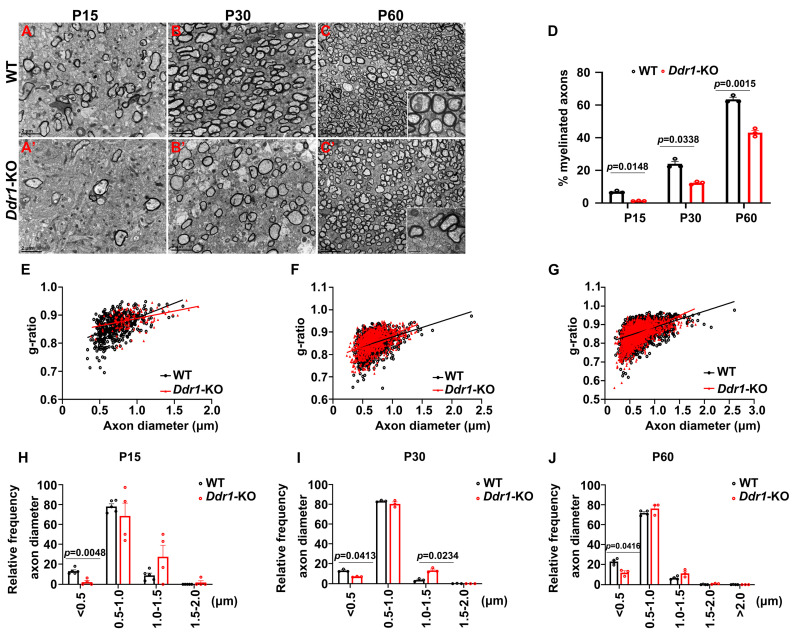
Knockout *Ddr1* altered the myelin structure and myelinated axon constitution in the corpus callosum. (**A**–**C’**) Electron microscopic images of the corpus callosum (CC) from wild-type and *Ddr1*-KO mice at indicated stages. Scale bars represent 2 µm. The insets show both typical myelinated axons and unmyelinated axons from two groups. The scale bar represents 0.5 µm. (**D**) Quantification of the percentage of myelinated axons in the corpus callosa from wild-type and *Ddr1* mutants at indicated stages. Error bars indicate means ± SEM. *n* ≥ 3. (**E**) The relationship between diameters and g-ratios of axons from the corpus callosa of wild-type and *Ddr1*-KO mice at P15. The averaged g-ratios were 0.8638 ± 0.001796 for wild-type mice (523 axons from 3 animals; indicated by black squares) and 0.8843 ± 0.002994 for *Ddr1*-KO mice (121 axons from 3 animals; indicated by red triangles); *p* < 0.0001. (**F**) The relationship between diameters and g-ratios of axons from the corpus callosa of wild-type and *Ddr1*-KO mice at P30. The averaged g-ratios were 0.8503± 0.001297 for wild-type mice (1096 axons from 3 animals; indicated by black squares) and 0.8567 ± 0.001406 for *Ddr1*-KO mice (936 axons from 3 animals; indicated by red triangles); *p* = 0.0008. (**G**) The relationship between diameters and g-ratios of axons from the corpus callosa of wild-type and *Ddr1*-KO mice at P60. The averaged g-ratios were 0.8522± 0.001174 for wild-type mice (1517 axons from 3 animals; indicated by black squares) and 0.8589 ± 0.001381 for *Ddr1*-KO mice (882 axons from 3 animals; indicated by red triangles); *p* = 0.0003. (**H**) Quantification of the percentage of myelinated axons by size in the corpus callosa of wild-type and mutant mice at P15. A decreased number of myelinated small-diameter axons (<0.5 µm) were observed in *Ddr1* mutants. (**I**) Quantification of the percentage of myelinated axons by sizes in the corpus callosa of wild-type and mutant mice at P30. (**J**) Quantification of the percentage of myelinated axons by sizes in the corpus callosa of wild-type and mutant mice at P60. A decreased number of myelinated small-diameter axons (<0.5 µm) were also observed in *Ddr1* mutants. Error bars indicate means ± SEM.

**Figure 5 ijms-24-10318-f005:**
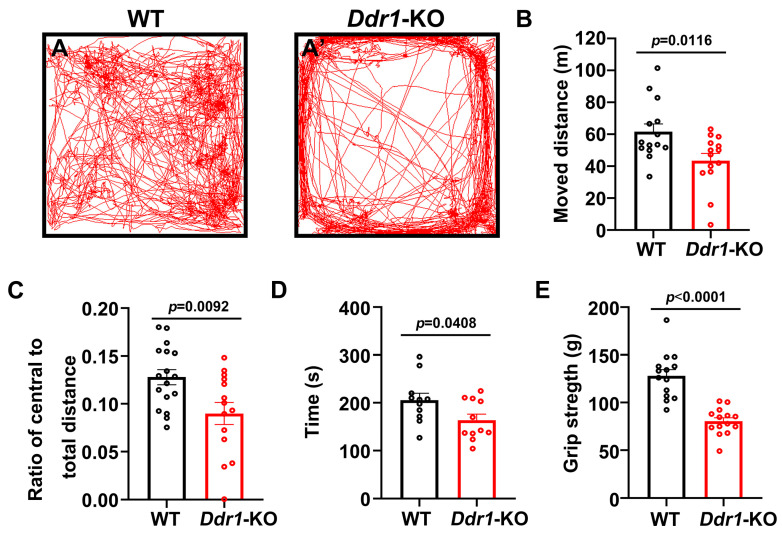
Deficiency of *Ddr1* impaired the motor function in mice. (**A,A’**) Locomotor activity of P60 wild-type mice (*n* = 14) (**A**) and *Ddr1*-KO mice (*n* = 14) (**A’**) in the open field test; (**B**) moved distance; (**C**) ratio of the central to total distances; (**D**) The latency time before falling from the rod of each mouse for wild-type mice (*n* = 14) or mutant mice (*n* = 14) at P60. Rotarod analysis showed a worse motor performance in *Ddr1*-KO mice; (**E**) forelimb grip strengths in wild-type mice (*n* = 11) and *Ddr1*-KO mice (*n* = 11) at P60, showing significantly reduced forelimb grip strength in the *Ddr1*-KO mice (*n* = 11) compare to in the wild-type mice (*n* = 11) at P60. Error bars indicate means ± SEM. Black or red dot represents individual data point.

**Figure 6 ijms-24-10318-f006:**
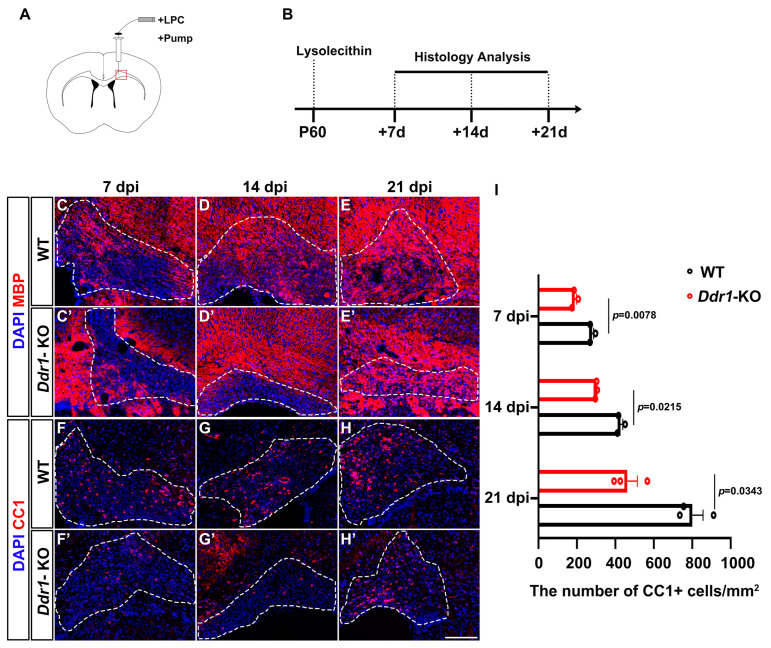
*Ddr1* mutation attenuated remyelination in LPC-induced demyelinated lesions. (**A**) LPC injection site in the corpus callosum of P60 adult mice brains. (**B**) Schedule of histological analyses in the LPC lesion paradigm. (**C**–**H’**) LPC lesions of wild-type and *Ddr1*-KO mice stained for MBP indicated in red (**C**–**E’**) or CC1 indicated in read (**F**–**H’**) at 7, 14, and 21 days post-injection and counterstained with DAPI (indicated in blue). Dashed lines delineate lesion areas. The scale bar represents 25 µm. (**I**) The numbers of CC1^+^ cells per mm^2^ in wild-type and *Ddr1*-KO mouse lesion regions at 7, 14, and 21 dpi. Error bars indicate means ± SEM.

**Figure 7 ijms-24-10318-f007:**
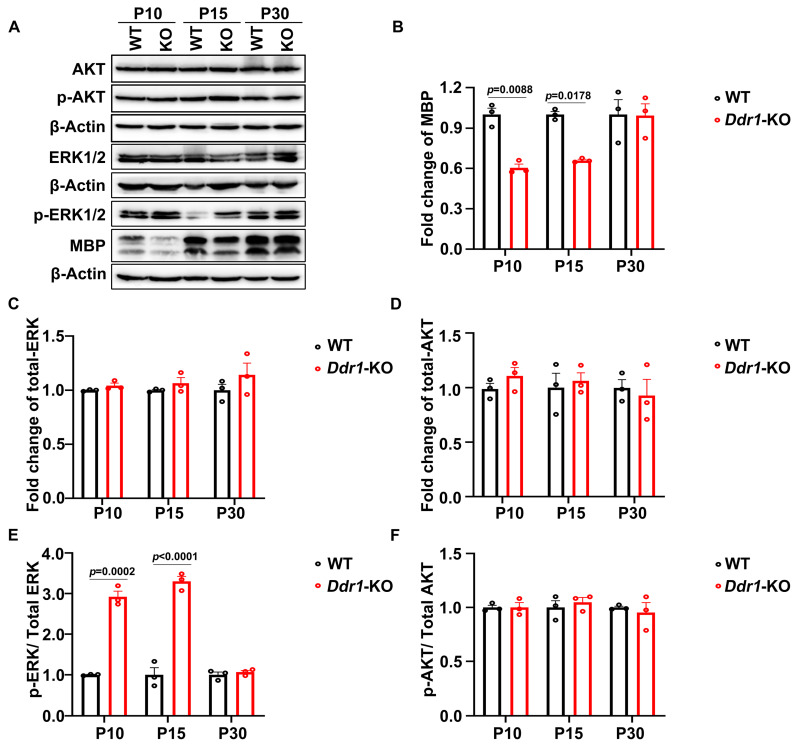
*Ddr1* deletion enhanced ERK activation. (**A**) The expression of phosphorylation of ERK, MBP, total ERK, total AKT, and phosphorylation of AKT in brainstem tissues from wild-type and *Ddr1*-KO mice at P10, P15, and P30. β-Actin was used as the internal control. (**B**) Quantification of MBP protein levels in the brainstem of *Ddr1-KO* mice which were normalized to those in the brainstem of the wild-type group. (**C**) Quantification of total ERK protein levels in the brainstem of *Ddr1-KO* mice which was normalized to those in the brainstem of the wild-type group. (**D**) Quantification of total AKT protein levels in the brainstems of *Ddr1-KO* and wild-type mice. (**E**) Quantification of ERK phosphorylation levels in total ERK in the brainstem tissues of *Ddr1*-KO and wild-type group. (**F**) Quantification of AKT phosphorylation levels in the total AKT in the brainstem tissues of *Ddr1-KO* and control group. Error bar indicates means ± SEM. *n* = 3.

## Data Availability

Data supporting the findings of this study shall be made available in the article and the Appendix A or from the corresponding authors upon reasonable request.

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
