# Peer review of "Evidence That DDR1 Promotes Oligodendrocyte Differentiation during Development and Myelin Repair after Injury"

_ijms, 2023, doi:10.3390/ijms241210318_

Round 1

Author Response

1.English language needs checking for spelling and grammar mistakes as well as logical coherence (see for instance l. 28 (development central nervous system), l. 52/234 (newly differential oligodendrocytes)). l. 168 (we ran these mice at a rotating bar), l. 183 (remyelination of the mice), l. 184 (nest), l.191, l. 242).

We carefully checked and revised the spelling, grammatical errors and logical coherence in the MS.

2.The alleged specific and selective expression of Ddr1 in CC1-, Myrf- and ASPA-positive NFOs (see Fig. 1 G,H,I) cannot be concluded from the depicted IHCs. In neither coimmuno-histochemical staining the red marker signal and the green Ddr1 signal are overlapping fully (Myrf) or even partially (CC1 and ASPA). The overlap for ASPA is as high as it is for the astrocytic marker GFAP. Probably this is due to bad staining quality but the deduced finding (l. 45-46, l. 58-61 and legend l. 68-70) cannot be proven by the data. Do the authors argue that Ddr1 expression is only expressed transiently before Myrf-positive cells start to express CC1? This would have to be proven by co-immunostainings at a later time point than P15. At least Ddr1 transcripts are still present at P21 and P30 when most Myrf-positive cells will be already in the stage of mature myelinating OLs. Therefore this explanation is probably not valid.

The issues raised by the reviewer are caused by the stage-specific expression of these markers in oligodendrocyte lineage and their differential cellular localizations. For instance, MYRF is localized in the nucleus while CC1 is located in cell membrane. In the white matter, ASPA is expressed in mature oligodendrocytes (OLs) much later than many other OL markers, including MBP, PLP and MYRF; and it is localized in cell bodies, nuclei and some processes. DDR1 is a receptor tyrosine kinase and located in cell membrane. In our manuscript, the majority of Ddr1-positive cells were co-stained with MYRF and only a small proportion of Ddr1+ cells were CC1+.

In Fig.1 it was a mistake that Ddr1-positive cells were co-stained with ASPA+ mature oligodendrocytes in the corpus callosum, and this is corrected in the revised MS. Thus, we think Ddr1 is selectively expressed in differentiating OLs.

3.From many other publications in the field and from the Fig. 2 I-L in this manuscript it is known, that at P7 and P10 there are already considerable numbers of positive cells for Myrf protein and PLP and Mbp transcripts present in the mouse corpus callosum. Therefore the near absence (one cell) of Ddr1-positive cells in Figure 1 B, C may be a hint for bad ISH quality on those tissue samples or argues against an expression in newly formed Myrf-positive OLs. In the same direction, the effect of the Ddr1 ko in mice seen at P10 (Fig. 2 I-L) in this case would precede the actual expression of Ddr1 in the corpus callosum. This would then argue for a non-cell autonomous effect of the Ddr1 ko and contradict the findings and conclusions of this manuscript.

Sorry for the possible confusions. The main reason why there are only a few Ddr1+ cells at P7 is that Ddr1-positive cells start to emerge in the external capsule and cingulum at this stage. In the revised figure, we used the more representative images to show the expression of Ddr1 in the corpus callosum.

4.The used mouse model is a full constitutive Ddr1 ko. Is there a phenotype of this ko? Do the mice behave and survive normally and have no other obvious organic defects caused by the deletion of Ddr1? It would be interesting if the detected effect is cell autonomous or if the deletion in non-oligodendroglial cell types could affect OL development. For example the reduced grip strength and motor functions as well could be caused by muscle weakness if Ddr1 is expressed in muscle tissue or peripheral nerves or changes in hormones such as testosterone or catecholamines. This should be ruled out. Are there expression data for those tissues or is an OL-specific conditional ko mouse line available?

Ddr1-KO mice appear to survive normally in the lab. Previous studies showed they have some abnormal aorta collagen fibril morphology. The authors have raised an important issue on the cell autonomous effect. Unfortunately, we can’t perform these recommended experiments due to the lack of Ddr1 conditional KO mouse line.

5.Absolute numbers or density of Sox10 and Olig2 cells, comparable to other markers in Fig. 2, would be helpful to evaluate the effect of the Ddr1 ko in the corpus callosum. Are there less, similar numbers or more oligodendroglial cells in the early stages of postnatal development? What is meant by “density” anyway? It is never defined in the legends or the methods how the cell density was calculated. Is it calculated per mm2 of the whole corpus callosum or is it only the cell number per slice? The same is true for other quantification such as the number of myelinated axons in Fig. 4. Is it calculated for the whole corpus callosum per section and somehow normalized to the counted area? It also is important at which position (rostral, medial or caudal relative to the Bregma) the quantifications were performed. More information regarding the methods and quantifications would be helpful.

As shown below, there are comparable numbers of Olig2+ oligodendroglial cells in the corpus callosum in both control and mutant mice at the early stages of postnatal development (P7).

As suggested, we calculated the number of cells per mm2 of the whole corpus callosum, and revised the description in the MS. Brain sections were collected consecutively from Bregma 1.1 mm to Bregma 1.34 mm of P10 or P15- P30 mouse brains. The description is included in the method as suggested.

In Fig 4, the brain sections were collected consecutively from Bregma 1.1 mm of P60 or P15-P30 mouse brains correspondingly. The description is added to the Method as suggested.

  1. Neither in the demyelination model nor in the behavioral testing paradigm the age of the tested/treated mice was stated in the figure, the legend or the methods section.

We apologize for the missing information. The descriptions are now added in the Legend and Method sections as suggested.

  1. In Fig. 5 the mice not only reduced their total travelled distance but also they avoided the center of the field, which also may be indicative for an anxiety-like phenotype. Did the authors analyze more aspects of changes in emotional or social behavior?

Thank you for pointing the possible anxiety-like phenotype out. We were not aware of this, and thus deleted this test data.

A good suggestion. Due to our lack of technical expertise, we did not analyze emotional or social behavior.

8.Why was the brain stem used to identify potential changes in signaling pathways despite the fact that all other experiments were performed on corpus callosum? There are several examples in the literature that there can be quite substantial differences in regulatory networks in oligodendrocytes from different CNS and even brain regions (for instance Olig1 dependence of OL maturation or mTOR signaling in oligodendroglial cholesterol biosynthesis in the brain and the spinal cord). Did the authors also perform marker analyses such as ISH or IHC quantification in the brain stem or the spinal cord to see if the effect would be comparable to that seen in the corpus callosum?

This is because OL differentiation occurs earlier in the brain stem (P0) than in the forebrain (around P4), which is better for study potential changes in signaling pathways.

Yes, we have performed Mbp and Plp1 ISH in the spinal cord (see below), and showed that the expression of Mbp and Plp1 were reduced from P0 to 4, but recovered at P7. Thus, Ddr1 mutation caused abnormal OLs differentiation in the spinal cord at early postnatal stages

9.Several publications on this topic show that Erk1/2 activation is positively regulating myelin thickness and OL differentiation also in remyelination events (also two recent ones cited in this manuscript (Gonsalvez et al and Tong et al.)). The argumentation that Ddr1 ko increases Erk activation and therefore leads to reduced OL differentiation and myelination is therefore contradictory and has to be discussed more in detail, if contradicting publications on this topic are cited. Also the title is then misleading since it indicates that Ddr1 is activating Erk signaling to promote oligodendrocyte differentiation.

Thanks for pointing out this error. The role of ERK1/2 signaling in OLs in the CNS is more complicated and conflict results have been reported. The complete picture of how the ERK proteins regulate myelination is still unclear. In our study, we found ablation of Ddr1 increased the Erk activation and led to abnormal myelination. We have revised the references and the title. The related papers (Suo, Na et al., 2019; Allan, Kevin C et al., 2021) are now included in the revised MS.

10.Was the quantification of Erk and Akt levels only performed on a sample size of 1 per genotype and condition as depicted in the Western blot? The shown Western blots especially for WT P10 do not look convincing to me, otherwise total Akt levels would be increased drastically at P10 in the ko tissue. Maybe the Figure could be modified to show all quantified Blots. A graph for total Akt and Erk levels would be helpful. In Fig. 7A there are also two rows labelled with ActB. One should be ActA, am I right?

Sorry for the possible confusions. For each statistical analysis in Fig7, the results were from independent experiments (see below AKT WB)

ACTB is also called β-Actin, and we have corrected the notation in the revised MS.

11.DDR1 by convention is normally used for the human protein orthologue. Ddr1 would be the mouse protein and Ddr1 in italics the mouse gene or transcript. The authors maybe should somehow conform their text and figures and do not use several different notations for the same protein or gene.

Thanks, it was corrected as suggested.

  1. The Supplementary figures S1 and S2 are missing and therefore their validity cannot be evaluated.

Sorry, they are now added in the revised MS.

Reviewer 2 Report

Others have shown DDR1 is present during in vitro oligodendrocyte differentiation (Silva et al., 2023) and may regulate peripheral axon diameter via myelin ensheathment in Drosophila (Corty et al., 2022). This manuscript provides in vivo evidence that DDR1 may be important for functional oligodendrocyte myelination i.e. normal motor function and may act through the ERK signaling pathway. 

The following major corrections are recommended prior to publication:

·         Figure 1.  G-L  Nuclear labeling would help to better distinguish individual cells and localization of cell markers. Additionally, the exposure time is too high for GFAP, Iba1, and NeuN.  In the corpus callosum of naïve mice astrocytes and microglia typically have fine cell processes and smaller cell bodies. The density of GFAP labeling is higher than expected for non-injured white matter. In I it is difficult to see dual immunolabeling of ASPA and DDR1.

·         The Supplementary data referenced throughout the manuscript was not provided and could not be reviewed. 

·         All graphs should include individual data points. 

·         Statistical Analysis of the data of WT vs DDR1-KO over time should be performed using Two-way ANOVA with post-hoc multiple comparisons.

·         Corpus callosum thickness is dependent on cell density, myelination, axon number and axon caliber and varies along the rostral-caudal dorsal-ventral regions. The ROI within the corpus callosum should be clearly defined with 1) coronal plane coordinates and 2) a diagram showing the ROI for each figure with quantification.

·         The electron microscopy images/data in figure 4

a.       Should include scale bars

b.       Should include higher magnification examples of unmyelinated axons with myelinated axons.

c.       D-F the data should be shown in one graph with two-way ANOVA to show significant changes in myelinated axons over time and between genotypes.

d.       J is missing data

e.       Axons < 0.3 ums are generally unmyelinated and should be excluded or quantified separately.

f.        Unmyelinated axons should be visible in A. Difficult to see axons in P15 Ddr1-KO image and may include more cytoplasm from adjacent cell than other images. 

g.  Unmyelinated axons should also be quantified.

·         Please explain why brainstem tissue was used the western blot analysis and not the corpus callosum ROI.

·         Western blot analysis of additional ERK signaling components would strengthen these findings.

·         Why is p-AKT and total AKT decreased in P15 WT mice?

·         Figure 6

a.       Quantification of CC1 in sham controls should be provided to show loss of oligodendrocytes.

b.       CC1+ cells in the cingulum should be excluded or quantified separately.  

c.       Myelination can not be shown with CC1+ cell quantification. Myelination is typically shown with images of myelin staining, IHC for myelin proteins, and/or EM. 

d.       Data should be shown in one graph with two-way ANOVA and post-hoc multiple comparisons.

·         The open field behavioral data shown is also associated with thigmotaxis or anxiety-related behavior in rodents.

Author Response

  1. Figure 1.  G-L  Nuclear labeling would help to better distinguish individual cells and localization of cell markers. Additionally, the exposure time is too high for GFAP, Iba1, and NeuN.  In the corpus callosum of naïve mice astrocytes and microglia typically have fine cell processes and smaller cell bodies. The density of GFAP labeling is higher than expected for non-injured white matter. In I it is difficult to see dual immunolabeling of ASPA and DDR1.

Good suggestions. As reported, MYRF is a nuclear protein which could serve to distinguish individual cells.

For some unknown reasons, astrocytes in the cingulum region appear to have a higher density of GFAP labeling. In l, we meant Ddr1-positive cells were not co-stained with ASPA, and now corrected this error in the revised manuscript.

2.The Supplementary data referenced throughout the manuscript was not provided and could not be reviewed.

We apologize, and they are now added in the result section.

3.All graphs should include individual data points.

We have revised all graphs as suggested.

4.Statistical Analysis of the data of WT vs DDR1-KO over time should be performed using Two-way ANOVA with post-hoc multiple comparisons.

It had been revised.

  1. Corpus callosum thickness is dependent on cell density, myelination, axon number and axon caliber and varies along the rostral-caudal dorsal-ventral regions. The ROI within the corpus callosum should be clearly defined with 1) coronal plane coordinates and 2) a diagram showing the ROI for each figure with quantification.

We had showed the ROI for each figure with qualification using white dashed lines, and brain sections were collected consecutively from Bregma 1.1 mm to Bregma 1.34 mm of the P10 or P15 -P30 mouse brains correspondingly.

  1. The electron microscopy images/data in figure 4
  2. Should include scale bars

It was added.

  1. Should include higher magnification examples of unmyelinated axons with myelinated axons.

As suggested, higher magnification representative images showing unmyelinated vs myelinated axons of P60 are inserted into Figure 6C and 6C’.

  1. D-F the data should be shown in one graph with two-way ANOVA to show significant changes in myelinated axons over time and between genotypes.

Revised as suggested.

  1. J is missing data.

The missing data may be cause by the software issue. Please see the fig below:

  1. Axons < 0.3 ums are generally unmyelinated and should be excluded or quantified separately.

We agree. Indeed, we excluded the axon < 0.3 um.

  1. Unmyelinated axons should be visible in A. Difficult to see axons in P15 Ddr1-KO image and may include more cytoplasm from adjacent cell than other images.

As suggested, we replaced the representative images in the figure 4A and 4A’.

  1. Unmyelinated axons should also be quantified.

We had quantified the unmyelinated axons and performed the percentage of myelinated axons at P15, P30 and P60 (Figure 4D-F).

  1. Please explain why brainstem tissue was used the western blot analysis and not the corpus callosum ROI.

This is because OL differentiation occurs earlier in the brain stem (P0) than in the forebrain (around P4), which is better for study potential changes in signaling pathways.

  1. Western blot analysis of additional ERK signaling components would strengthen these findings.

This is a good suggestion. However, we could not obtain good antibodies for additional ERK signaling.

  1. Why is p-AKT and total AKT decreased in P15 WT mice

Sorry for the possible confusions. This could be caused by some technical issues. p-AKT and total AKT statistical analyses were from 3 independent animals, and showed some variations (see below AKT WB)

We added the graph for total Akt and Erk levels in the Fig7 to help assess the protein level.

  1. Figure 6
  2. Quantification of CC1 in sham controls should be provided to show loss of oligodendrocytes.

This is a good suggestion. We compared the remyelination between wt and Ddr1-KO mice in LPC-induced demyelinated lesions, so we did not setup sham control. And it had been demonstrated that there was no loss of oligodendrocytes in sham controls. (Wang, Yao et al. “Regulatory T cells alleviate myelin loss and cognitive dysfunction by regulating neuroinflammation and microglial pyroptosis via TLR4/MyD88/NF-κB pathway in LPC-induced demyelination.” Journal of neuroinflammation vol. 20,1 41. 18 Feb. 2023, doi:10.1186/s12974-023-02721-0)

b.CC1+ cells in the cingulum should be excluded or quantified separately.

We only qualified the CC1+ cells in the lesion area, which is outlined area with a higher density of nuclei (DAPI, blue).

  1. Myelination can not be shown with CC1+ cell quantification. Myelination is typically shown with images of myelin staining, IHC for myelin proteins, and/or EM.

Agree, and we used MBP immunofluorescence (red) and DAPI nuclear staining in brain sections to show the myelination (Fig6 C-E).

  1. Data should be shown in one graph with two-way ANOVA and post-hoc multiple comparisons.

Revised as suggested.

  1. The open field behavioral data shown is also associated with thigmotaxis or anxiety-related behavior in rodents.

Thank you for pointing it out. We were not aware of this, and thus deleted this test data.

Reviewer 3 Report

The manuscript have informations related to DDR1KO in mouse. 

Introduction should be more explicative of the current state of art, and in the discussion data should be considered with respect to currectly available data. The litterature to which it refers is too old new data are available see review Vilella E, Gas C, Garcia-Ruiz B, Rivera FJ. Expression of DDR1 in the CNS and in myelinating oligodendrocytes. Biochim Biophys Acta Mol Cell Res. 2019 Nov;1866(11):118483. doi: 10.1016/j.bbamcr.2019.04.010. Epub 2019 May 18. PMID: 31108116.

Majour points: 

the author stated that

In this study, we report that DDR1 is selectively upregulated in oligodendrocytes during differentiation and myelin formation stages”

Discussion: “Line 228-230 : little was known about its in vivo role in oligodendrocyte development. In this study, for the first time, we provided the genetic and molecular evidence that Ddr1 regulates the tempo of OL differentiation and myelination and influences the motor function of animals.”

The following reviewExpression of DDR1 in the CNS and in myelinating oligodendrocytes must be cited, data reported overlap with previous studies. Authors must highlight and missing points, in the introduction, especially stressing which are the novelty of the presented study compared to previous studies: Vilella E, Gas C, Garcia-Ruiz B, Rivera FJ. Expression of DDR1 in the CNS and in myelinating oligodendrocytes. Biochim Biophys Acta Mol Cell Res. 2019 Nov;1866(11):118483. doi: 10.1016/j.bbamcr.2019.04.010. Epub 2019 May 18. PMID: 31108116.

Overstatement of their results are also in the text

Lines 40-43“Recently, DDR1 was shown to be enriched  in the murine transcriptome of OL lineage and post-mortem samples of human cerebral cortex [13, 14]”

References 13 and 14 refer to article of 2010 and 2014. They are not so recent. Maybe some refences is missing? Also because in the abstract it is mentioned that “mounting studies have demonstrated receptor tyrosine kinases (RTKs..”), BUT no refences for this are given in the introduction. Please reformulate or add missing information.

Lines 233-235 “In the present study, we discovered that Ddr1 is predominantly expressed in the newly differential OLs in early postnatal CNS when oligodendrocytes undergo active differentiation and myelination (Figure 1 and Figure S1). The strong expression of Ddr1 in differentiating OLs appears to be important for regulating OL differentiation and myelin 236 formation.”

It was previously reported that : “In vivo and in vitro data suggest that DDR1 expression is upregulated during the transition from OPCs to OLs from Vilella et al., 2019) , therefore this is not a discovery of the authors

Please reformulate or add missing information.

Minor points

The letter G in figure 4 has different size compared to the others in figure

Citation 13 and several others does not look correct for IJMS standards

13. Zhang, Y.; Chen, K.; Sloan, S. A.; Bennett, M. L.; Scholze, A. R.; O'Keeffe, S.; Phatnani, H. P.; Guarnieri, P.; Caneda, 404 C.; Ruderisch, N.; Deng, S.; Liddelow, S. A.; Zhang, C.; Daneman, R.; Maniatis, T.; Barres, B. A.; Wu, J. Q., An RNA-405 sequencing transcriptome and splicing database of glia, neurons, and vascular cells of the cerebral cortex. The Journal of neuro-406 science : the official journal of the Society for Neuroscience 2014, 34 (36), 11929-47.

Should be: Zhang Y, Chen K, Sloan SA, Bennett ML, Scholze AR, O'Keeffe S, Phatnani HP, Guarnieri P, Caneda C, Ruderisch N, Deng S, Liddelow SA, Zhang C, Daneman R, Maniatis T, Barres BA, Wu JQ. An RNA-sequencing transcriptome and splicing database of glia, neurons, and vascular cells of the cerebral cortex. J Neurosci. 2014 Sep 3;34(36):11929-47. doi: 10.1523/JNEUROSCI.1860-14.2014. Erratum in: J Neurosci. 2015 Jan 14;35(2):846-6. PMID: 25186741; PMCID: PMC4152602.

Author Response

  1. the author stated that

In this study, we report that DDR1 is selectively upregulated in oligodendrocytes during differentiation and myelin formation stages”

Discussion: “Line 228-230 : little was known about its in vivo role in oligodendrocyte development. In this study, for the first time, we provided the genetic and molecular evidence that Ddr1 regulates the tempo of OL differentiation and myelination and influences the motor function of animals.”

The following reviewExpression of DDR1 in the CNS and in myelinating oligodendrocytes must be cited, data reported overlap with previous studies. Authors must highlight and missing points, in the introduction, especially stressing which are the novelty of the presented study compared to previous studies: Vilella E, Gas C, Garcia-Ruiz B, Rivera FJ. Expression of DDR1 in the CNS and in myelinating oligodendrocytes. Biochim Biophys Acta Mol Cell Res. 2019 Nov;1866(11):118483. doi: 10.1016/j.bbamcr.2019.04.010. Epub 2019 May 18. PMID: 31108116.

Overstatement of their results are also in the text

As suggested, we cited this reference and described the novelty of the presented study compared to the previous studies.  

In this review, the authors have done an excellent work in reviewing the previous work on DDR1 isoform and expression, and DDR1 mutations in cancers. Although Ddr1 expression in oligodendrocyte linage was also introduced, most references that were cited in this review are RNA-seq and In Vitro data. Although RNA in situ hybridization showed that Ddr1 is expressed after myelination, we showed that DDR1 is expressed earlier in differentiating and newly differentiated OLs. In addition, there was no study on the functional involvement of Ddr1 in oligodendrocyte development and myelinogenesis.

  1. Lines 40-43“Recently, DDR1 was shown to be enriched in the murine transcriptome of OL lineage and post-mortem samples of human cerebral cortex [13, 14]”

References 13 and 14 refer to article of 2010 and 2014. They are not so recent. Maybe some refences is missing? Also because in the abstract it is mentioned that “mounting studies have demonstrated receptor tyrosine kinases (RTKs..”), BUT no refences for this are given in the introduction. Please reformulate or add missing information.

Thanks for pointing out these issues, the related papers (31-34) are now cited in the discussion section and included in the reference list.

  1. Lines 233-235 “In the present study, we discovered that Ddr1 is predominantly expressed in the newly differential OLs in early postnatal CNS when oligodendrocytes undergo active differentiation and myelination (Figure 1 and Figure S1). The strong expression of Ddr1 in differentiating OLs appears to be important for regulating OL differentiation and myelin 236 formation.”

It was previously reported that : “In vivo and in vitro data suggest that DDR1 expression is upregulated during the transition from OPCs to OLs from Vilella et al., 2019) , therefore this is not a discovery of the authors

Please reformulate or add missing information.

Thanks for pointing out this error, and we corrected this writing error.

  1. The letter G in figure 4 has different size compared to the others in figure

Sorry for the oversight. It was corrected.

  1. Citation 13 and several others does not look correct for IJMS standards
  2. Zhang, Y.; Chen, K.; Sloan, S. A.; Bennett, M. L.; Scholze, A. R.; O'Keeffe, S.; Phatnani, H. P.; Guarnieri, P.; Caneda, 404 C.; Ruderisch, N.; Deng, S.; Liddelow, S. A.; Zhang, C.; Daneman, R.; Maniatis, T.; Barres, B. A.; Wu, J. Q., An RNA-405 sequencing transcriptome and splicing database of glia, neurons, and vascular cells of the cerebral cortex. The Journal of neuro-406 science : the official journal of the Society for Neuroscience 2014, 34 (36), 11929-47.

Should be: Zhang Y, Chen K, Sloan SA, Bennett ML, Scholze AR, O'Keeffe S, Phatnani HP, Guarnieri P, Caneda C, Ruderisch N, Deng S, Liddelow SA, Zhang C, Daneman R, Maniatis T, Barres BA, Wu JQ. An RNA-sequencing transcriptome and splicing database of glia, neurons, and vascular cells of the cerebral cortex. J Neurosci. 2014 Sep 3;34(36):11929-47. doi: 10.1523/JNEUROSCI.1860-14.2014. Erratum in: J Neurosci. 2015 Jan 14;35(2):846-6. PMID: 25186741; PMCID: PMC4152602.

Sorry for the oversight. This part was revised

Round 2

Author Response

1) There are again some minor mistakes in the written text, especially concerning singular and plural use of substantives (l. 43 the cells of oligodendrocyte lineage, l.116: the development of myelin tract; l.26-27: myelin sheath, a major component of the white matter, is). For “myelin tract” and “myelin sheath” the plural form would make more sense.an it rather would be “cells of the oligodendrocyte lineage”.

**We checked these mistakes and revised.

2) The labelling of the Figures should be adjusted to a readable font size in every figure. Especially in Figure 2 the y-axis labelling and the WT/Ddr1 KO annotations are very small and difficult to read.

** We have revised all graphs as suggested.

3) The authors provided western blot results for AKT and p-AKT in the postnatal brainstem that do not at all look like the ones shown in Fig. 6A of the manuscript. Since two referees questioned the reduced AKT levels at P15 in WT mice compared to P10 and P30 this Figure should be exchanged with a new Western blot image showing the actual expression levels used for quantification.

** We have exchanged the Fig with new WB image for AKT and p-AKT with actin.

4) Unfortunately, it still seems to me that the authors should be more careful in writing/revising their manuscript since in their main findings they contradict themselves. In the abstract they claim that Ddr1 is needed for timely developmental myelination and remyelination and that it performs its function by increasing Erk1/2 signaling (l. 16-18: In this study, we report that Ddr1 is selectively upregulated in newly differentiated oligodendrocytes in early postnatal CNS, and regulates oligodendrocyte differentiation and myelination by stimulating ERK pathway.). Inconsistent with this argumentation, in the results section they show that Erk1/2 phosphorylation and therefore Erk downstream signaling is induced by a knockout of Ddr1 concomitant with impaired myelination and only cite publications that annotate Erk1/2 signaling with reduced myelination and OL differentiation. The authors should carefully read their manuscript and revise it for better logical coherence. Also Erk1/2 functions are published to be ambivalent during OL differentiation, as the authors also mentioned in their rebuttal letter. In my opinion this has to be discussed. It would be better to not only cite the publications that match your data best, but say that there are different data concerning the role of Erk1/2 signaling in OL differentiation and discuss how their data fit into the recent literature.

** It had been improved in the discussion part.

5) The results of the open field test do not have to be removed since they show an additional phenotype of Ddr1 ko in mice. The authors just have to state that they also could be a sign for anxiety-like behavior. Additionally non-cell-autonomous phenotypes due to constitutive KO could at least be mentioned in the discussion.

** It had been revised in the discussion part.

6) Still the initial age of the mice used for LPC-injection is not stated anywhere. It is just stated how many days after induction they sacrificed the mice.

**It is now added in the revised MS

Reviewer 3 Report

I think discussion could have been improved

Author Response

1) I think discussion could have been improved

** It had been improved.